# Cell Count Differentials by Cytomorphology and Next-Generation Flow Cytometry in Bone Marrow Aspirate: An Evidence-Based Approach

**DOI:** 10.3390/diagnostics13061071

**Published:** 2023-03-11

**Authors:** Rafael Ríos-Tamayo, María José Sánchez, Sandra Gómez-Rojas, Miguel Rodríguez-Barranco, Gloria Pérez Segura, Daniel Redondo-Sánchez, Gonzalo CARREÑO-TARRAGONA, Antonio Rodríguez Nicolás, Francisco Ruiz-Cabello, Pilar Jiménez, Rafael Alonso, Juan José Lahuerta, Joaquín Martínez-López, Rafael F. Duarte

**Affiliations:** 1Fundación para la Investigación Biomédica del Hospital Universitario Puerta de Hierro-Majadahonda, Hematology Department, Hospital Universitario Puerta de Hierro, 28222 Majadahonda, Spain; 2Centro de Investigación Biomédica en Red de Epidemiología y Salud Pública (CIBERESP), 28029 Madrid, Spain; 3Instituto de Investigación Biosanitaria de Granada (Ibs.GRANADA), 18014 Granada, Spain; 4Registro de Cáncer de Granada, Escuela Andaluza de Salud Pública (EASP), 18080 Granada, Spain; 5Department of Preventive Medicine and Public Health, University of Granada, 18016 Granada, Spain; 6Hematology Department, Hospital Universitario 12 de Octubre, Instituto de Investigación del Hospital Universitario 12 de Octubre, 28041 Madrid, Spain; 7Immunology Department, Hospital Universitario Virgen de las Nieves, 18014 Granada, Spain

**Keywords:** bone marrow aspirate, cytomorphology, next generation flow, differential cell counts, 200 vs 500 cutoffs, ISO15189, preanalytical quality, multiple myeloma, plasma cells, blasts, diagnostic efficiency, clinical laboratory, clinical significance

## Abstract

Despite a lack of evidence, a bone marrow aspirate differential of 500 cells is commonly used in the clinical setting. We aimed to test the performance of 200-cell counts for daily hematological workup. In total, 660 consecutive samples were analyzed recording differentials at 200 and 500 cells. Additionally, immunophenotype results and preanalytical issues were also evaluated. Clinical and statistical differences between both cutoffs and both methods were checked. An independent control group of 122 patients was included. All comparisons between both cutoffs and both methods for all relevant types of cells did not show statistically significant differences. No significant diagnostic discrepancies were demonstrated in the contingency table analysis. This is a real-life study, and some limitations may be pointed out, such as a different sample sizes according to the type of cell in the immunophenotype analysis, the lack of standardization of some preanalytical events, and the relatively small sample size of the control group. The comparisons of differentials by morphology on 200 and 500 cells, as well as by morphology (both cutoffs) and by immunophenotype, are equivalent from the clinical and statistical point of view. The preanalytical issues play a critical role in the assessment of bone marrow aspirate samples.

## 1. Introduction

Despite major advances in the diagnosis of blood diseases (BD), bone marrow (BM) examination remains a key step in the diagnostic workup in most hematological malignancies (HM) and many non-tumor diseases [1]. The cytomorphological study of the BM aspirate (BMA) is carried out using an optical microscope to analyze BMA smears stained with a panoptic stain. A nucleated differential cell count (DCC) is performed to estimate the proportion for each cell type and to calculate the myeloid to erythroid ratio (M/E r).

Diagnosis is the most difficult stage of clinical activity. The diagnostic process of BD requires a comprehensive evaluation of clinical and epidemiological data, including family and personal history. Comorbidities, as well as previous and current therapies, must be indicated in the electronic medical records, with free access to clinical analytics, imaging, and any other complementary test. A complete blood count and biochemistry is mandatory, complemented by a peripheral blood (PB) smear cytomorphological analysis when indicated. The BMA sample is commonly used for several complementary studies including cytomorphology, cytochemistry, flow cytometric immunophenotyping (IP), cytogenetics, and molecular analysis. Additionally, a BM trephine biopsy is performed when needed. Each of these techniques have potential preanalytical (PA), analytical, and postanalytical quality events. It is well known that the PA phase is the source of most errors in the clinical laboratory. However, little is known about the impact of PA in BMA DCC.

Nowadays, the diagnostic workup in the hematology setting needs a sequential, multidisciplinary, and coordinated approach, to reach the right diagnosis, in the shortest possible time, using the least number of tests possible, at the lowest cost, with the minimal impact on the health-related quality of life. This dynamic process is recently called integrated diagnosis. Indeed, safety, accuracy, standardization, and efficiency are the basis of modern hematologic diagnosis.

BMA is considered a safe procedure, with a low risk of morbidity. Life-threatening complications are eventually reported but mortality is unusual. Major advances have been made using PB in order to avoid BMA, but there is still a long way to go until BM can be replaced by PB. Bioinformatics and artificial intelligence will help to achieve this goal. In the meantime, a BMA DCC remains an essential step in the daily hematologic workup, for the diagnosis, prognosis, follow-up, and response evaluation of most BD. Taking into account its importance, the lack of standardization and the controversy about the number of cells to be counted is striking. Even the information about normal BMA DCC is scarce and new data are welcomed in this regard [2,3].

Based on expert opinions (EO), the most frequently recommended cell count for BMA DCC is 500-cell (range, 200–1000), depending on the diagnostic question at hand. The most relevant recommendations about BMA DCC [4,5,6,7,8,9,10,11,12,13,14] are summarized in Table 1. Although some organizations, such as the International Council for Standardization in Hematology (ICSH) or the World Health Organization (WHO), have recommended 500-cell count, this is not validated by an evidence-based approach. Several prominent authors state that a 200- to 500-cell count is generally adequate, but again, this is based only on EO. Therefore, controversy may lead to difficulty on clinical decision making in the clinical laboratory setting, mainly due to a lack of standardization.

Our laboratory is accredited by the ISO 15189 Standard [15], with the BMA DCC included in the technical scope. Our specific standard operating procedure (SOP) for this technique states that a minimum of 200-cell count should be analyzed. Consequently, higher counts could be needed for difficult cases or when the proportion of a particular cell of interest is near of a clinical cutoff that separates different entities.

The first evidence-based approach to compare two levels of DCC was reported recently, analyzing 300- versus (vs) 500-cell count in 165 cases, as part of routine patient care, demonstrating a lack of statistically and clinically meaningful differences between both methods [14]. To the best of our knowledge, this study has not been replicated or validated. We therefore aimed to, firstly, compare two techniques (cytomorphology vs IP) for BMA DCC, secondly, to contrast two levels of DCC (200- vs 500-cell DCC), and finally, to assess the impact of PA events (PAE) in the daily clinical practice of a hematologic clinical laboratory accredited by the ISO 15189 Standard.

## 2. Materials and Methods

All consecutive BMA samples collected as part of routine patient care between August 2019 and August 2020 in our clinical laboratory were prospectively included in the study.

The BMA sample was taken according to our BM puncture–aspiration SOP. In brief, the requesting physician must inform the patient and sign the specific informed consent (IC) form to perform the technique. Additional IC are needed for cytogenetics and molecular studies. Local anesthesia was used in the puncture site, usually the upper third of the sternum, and 5–20 cc of BMA were taken in one or more aspirations. The samples were immediately transferred to a previously labeled tube with ethylenediaminetetraacetic acid anticoagulant, which is sent to the sample reception area of the lab in less than four hours, together with the corresponding request. A primary tube is used for each complementary test whenever possible.

BMA DCC was performed according to our BM cytomorphologic study (BMCS) and BM cytochemistry SOPs. The laboratory technician checks the concordance between the request and the sample at the time of receipt, as well as the characteristics of the sample. Any PAE is recorded in a standardized manner to be later included in the BMCS report. The presence of a clot in the specimen may potentially alter results so a detailed visual examination is mandatory. No sample is rejected for any reason, but clotted samples (CS) must be interpreted with caution. CS and other PAEs were analyzed in samples in which a 500-cell DCC was reached (group A) and in those in which that count could not be achieved (group B). Six BM smears are prepared for each sample of which two are stained with a panoptic stain, preferably May–Grünwald–Giemsa. In accordance with our quality system, the BMCS report must be issued within 48 h after receipt of the sample.

The DCC was implemented by the hematologist in charge. All cases were handled as part of routine clinical laboratory evaluations, using a standard digital optical microscopy (Leyca, DMD-108) and a manual cell counter giving percentages only as whole numbers. The requests forms must include demographic data, suspected or confirmed diagnosis, and current indication of the study. The first step was confirming the reason of the request, assessing the clinical context on the digital clinical history. After a preliminary evaluation (pre-count), global assessment of quality of the smear is recorded, including the type of study (new or re-evaluation), type of patient (adult or pediatric), the visible particles amount, cellularity (CE), megakaryocytes (ME), cell distribution, cell monomorphism, presence of cell clusters or syncytia, and M/E r. CE and ME were classified in three categories (high, normal, and low). Very low-CE samples were in turn subclassified into desert samples (DS) or semi-DS (SDS) when the higher count reached was 50 or 100, respectively. Standard morphologic criteria were used for the microscopic identification of each cell type. DCCs were recorded at the 200-cell mark and then the 500-cell, including the percentage of main cells: blasts (BL), promyelocytes (PR), myelocytes, metamyelocytes, bands, neutrophils, eosinophils, basophils, erythroblasts (ER), lymphocytes (L), monocytes (M), and plasma cells (PC). Then, a final cell evaluation (post-count) was made in order to assess significant dysplasia in any cell line, macrophages characteristics, and confirm the DCC. Iron staining was requested if myelodysplastic syndrome (MDS) was suspected. Myeloperoxidase, esterase, and periodic acid–Schiff cytochemical studies were used to confirm the lineage of acute leukemia (AL).

Since 2011, as part of the ISO15189 Standard accreditation for BMCS, a quarterly internal quality control is used to estimate interobserver variability among the section staff. In addition, a twice a year participation on the external quality control of the American College of Pathologist is mandatory.

A parallel BMCS study in an independent hematology department of other tertiary center was used as a control group.

For IP, BM samples were stained for cell surface markers using a direct immunofluorescence technique. Eight-color combinations of monoclonal antibodies (Mab) were used to identify the different cells subsets. The specificity and fluorochromes of each reagent used are listed in Appendix A. Stained cell suspensions were analyzed on a FACSCanto II flow cytometer (BD Biosciences, San José, CA, USA). An average of 30,000 events per tube corresponding to the whole cellularity was acquired. The InfinicytTM22.0 software (Cytognos S.L., Salamanca, Spain) was employed for multiparametric analysis. For instrument setup, BD one flow set up SOP were used. All Mabs were purchased from BD Biosciences, San Diego, CA. Flow cytometric differential counts (FDC) were reported in most cases only for specific types of cells. Only cells and percentages included in the final IP clinical report were used for evaluations in this study.

Comparisons of cells percentages were made between 200-cell and 500-cell DCCs BMCS, 200-cell and IP, and 500-cell and IP. Correlation coefficients (r) and linear regression statistics were estimated with SPSS package, v. 20 (IBM Corp., Armonk, NY, USA). In addition, Deming regression was performed to compare the two data sets using R, v.4.0.2 [16] and mcr v.1.2.1 [17]. Moreover, 2 × 2 contingency tables were created using 500-cell DCC as the gold standard to assess the impact of a 200-cell DCC on disease classification in relation to myeloblasts, for 5%, 10%, and 20% cutoffs. Sensitivity (Se), specificity (Sp), positive and negative predictive values (PPV, NPV) were calculated.

The 200-cell DCC was considered acceptable for daily clinical use if statistically significant differences between the two methods, and the two levels of DCC, were not founded, and no clinically significant diagnostic discrepancies could be demonstrated, particularly in BL or PC.

## 3. Results

In total, 660 consecutive samples were included in the study, in two sets of 330 samples each (training and validation sets); 244 cases were new and 416 (63%) follow-up studies; and 81 (12.3%) cases were pediatric. The complete set was compared with a control group (*n* = 122).

The characteristics of the complete set are indicated in Table 2. In 110 samples (16.7%, group B), a DCC of 500-cell could not be reached and they were therefore excluded for the comparison study. The remaining 550 samples formed the basis of the study, with 60.7% of them corresponding to reevaluation studies. The control group showed similar features, but the percentage of pediatric samples was significantly lower (12.3 % vs 5.7%, *p* = 0.035).

The final diagnosis for the complete set and the control group is displayed in Appendix A. Acute myeloid leukemia (AML) was the most frequent diagnosis in both the complete set and the control group, followed by multiple myeloma (MM).

PAEs are shown in Table 3. CS and any other PAE were analyzed separately. In the training set, both CS (1.4% vs 18.6%, *p* < 0.000) and any other PAE (11.1% vs 27.9%, *p* < 0.000) were significantly higher in the group B. The same is true for the complete set. In addition, 36 of 660 samples (5.5%) were CS, 19 in group A (3.5%) and 17 in group B (15.6%) (*p* < 0.000). The presence of any other PAE was detected in 128 of the remaining 623 (one sample was not received) non-CS (19.4%).

CS were identified in 5.5%, with this percentage being significantly lower in the control group (*p* = 0.027). CS were not associated with the type of study (new vs reevaluation) or the type of patient (adult vs pediatric). Regarding diagnosis, there was also no statistically significant differences, but non-Hodgkin lymphomas and multiple myeloma represent 1.2% and 1.1% of total CS (5.5%). With respect to CE, 31 (86%) out of 36 CS were hypocellular (*p* < 0.000): 16 low, five very low, eight SDS, and two DS.

The presence of any other PAE in non-CS was identified in 19.4%, with a similar value in the control group. Peripheral blood contamination (diluted samples) and low volume were significantly higher in the control group, but both PAEs are difficult to ascertain due to a lack of standardized criteria. The occurrence of platelet aggregates was detected in 99 cases (15%), being the most common non-CS PAE. Fifty-seven (57.6%) out of the 99 cases were hypocellular: 42 low, seven very low, five SDS, and three DS. A 500-cell DCC could not be reached in many of these samples.

Table 4 displays the r comparison analysis between both cutoff points for DCC (200-cell vs 500-cell) and both methods (BMCS DCC and IP FDC) for the six more relevant types of cells (BL, PR, ER, L, PC, and M) (*p* < 0.000, for all of them). Appendix A summarizes the linear regression analysis between both cutoff points and both methods. Figure 1 shows Deming linear regression and bias plots for BL and PC, provided that AML and MM are the two most frequent entities. Table 5 shows the Deming analysis and bias between both cutoffs. The linear regression analysis is highlighted in Appendix A, respectively. Except for the well-known difference between BMCS and IP for ER due to the IP method, the r for all other cell types is very high. No significant difference was detected between 200-cell and 500-cell DCC with respect to FDC.

The diagnostic performance of 200-cell vs 500-cell DCC (gold standard) for BL is shown in Table 6, Table 7 and Table 8. At 20% cutoff, the 500-cell DCC identified 20 of 550 cases with ≥20% blasts, whereas the 200-cell DCC detected 19 of them with ≥20% blasts and one case with <20%. The single discrepant case was a new myelomonocytic AML in which 18% and 20% blasts were detected by 200-cell and 500-cell DCC, respectively. In addition to blasts, more than 40% of the CE corresponded to monocytic lineage, both mature and precursor cells. Therefore, this minimal blast discrepancy count was not potentially correlated with a diagnostic discrepancy. In the same way, at 10% cutoff, 22 cases showed ≥10% blasts with both cutoffs and only one discrepant case had ≥10% (11%) blasts with 200-cell and <10% (9.5%) with 500-cell count. This patient had a de novo myelodysplastic syndrome (MDS), refractory anemia with excess blasts type 2. Finally, at 5% cutoff, among 30 cases with ≥5% blasts according with 500-cell DCC, 28 of them also had ≥5% blasts with 200-cell DCC whereas two cases had <5%. Both discrepant patients had de novo MDS, with 5% and 8% at 500-cell and 4% and 2% at 200-cell DCC, respectively. On the other hand, another two MDS cases had ≥5% blasts with 200-cell and <5% with 500-cell DCC, showing 5% and 6% at 200-cell and 4.5% and 4% at 500-cell DCC, respectively. Overall, only six cases (1.09%) among 550 were discrepant in any cutoff, with median of absolute differences of 1.75% (0.5–6%). The accuracy of 200-cell vs 500-cell DCC (gold standard) was 99%. For the established BL cutoffs of 20%, 10%, and 5%, Se was 95%, 100%, and 93.3%, whereas Sp was 100%, 99.8%, and 99.6%.

Regarding PCs at the cutoff of 10%, no discrepancies at all were found, yielding values of Se, Sp, PPV, and NPV of 100% for the performance of 200-cell vs 500-cell DCC.

Overall, this approach aims to achieve a simple, practical, and efficient cytomorphological contribution basis to the building of a comprehensive hematological diagnosis. The expertise of the cytomorphologist and the existence of a continuous quality improvement framework are key to reaching this.

## 4. Discussion

Despite major advances in the diagnosis of BD and attempts to automate the BMA DCC [18], BMCS remains a mandatory step in the diagnostic workup in most cases, particularly in the HM setting. BMA DCC has been used in daily clinical practice for many decades and has been the basis for countless studies. In spite of its relevance in both clinical trials and real-life, it is not yet completely standardized. Moreover, controversy persists in some critical points due to a remarkable lack of evidence. A 500-cell DCC has been considered the gold standard. However, this statement is based on individual expert opinions or recommendations by groups of experts integrated in some organizations. The lack of robust evidence-based support is striking. The first evidence-based study aimed at finding the right DCC for the routine patient care was reported recently, comparing 300-cell vs 500-cell DCC and suggesting that lowering routine DCCs from 500-cell to 300-cell would not affect quality of care. Nonetheless, this study has not been validated until now, it had a relatively small sample size, it did not include a control group, it did no compare BMCS DCC with IP, and it did not add information regarding PAEs. Our prospective study aims to answer all these questions, focusing on the routine patient care, but also taking into account the importance of standardizing the SOP as much as possible in order to comply with the ISO 15189 Standard.

Overall, our study not only reinforces the data obtained by Abdulrahman AA et al. [14] in Atlanta, but also suggests that a 200-cell DCC could be sufficient in the vast majority of studies for daily clinical practice without affecting the quality of the results. In fact, our performance analysis for the three blast cutoffs and the absolute differences found in discrepant cases are practically identical to the above indicated study. Our data also are consistent with previous recommendations for 200–500-cell DCC [4,5,9,13] and our own SOP states that a minimum of 200-cell DCC count should be analyzed. Nonetheless, in difficult cases with the cell of interest near an established cutoff, either de novo or reevaluations, particularly for the target of 5% blasts, a 500-cell or even higher DCC may be needed, but this occurs only in 1% of total studies.

On the other hand, little is known about the impact of PAEs in BMA DCC. The preanalytical phase of the total testing process is responsible for the majority of potential errors in the clinical laboratory. This phase involves ordering, patient preparation, specimen collection, identification, transport, and storage. Obtaining a BMA quality sample is of the utmost importance. There are no specific rejection criteria for BMA samples except for the general rule or rejection in cases of unidentified or misidentified samples, which were not identified in the study. BM clots faster than peripheral blood so it is not surprising that the percentage of coagulated samples is higher. In our study 5.5% where CS and 19.4% of non-CS had another PAE. Outside of CS, the list and the criteria for the PAEs for BMA DCC remain to be established in order to standardize their frequency and clinical impact.

The ICSH stated that FDCs should not be used as surrogate for DCCs, since they are complementary methods giving different information and their degree of correlation varies greatly. However, the standardization of next-generation flow has improved the accuracy and comparability of FDC. Results of IP FDC should be correlated with BMCS DCC. Nonetheless, few studies have assessed the correlation between DCC and FDC in the real-life setting. It is well known the impact of the ER-lysis method on the ER count by FDC. With this logical exception, the correlation for the other types of cells between DCC and FDC in our study is remarkably high.

Our study has several strengths. It is a prospective study throughout one year, with two consecutive set of samples, carried out in an ISO 15189 accredited laboratory with BMA DCC included in the technical scope since 2011. The global sample size is high, and an independent control group is included. The study points out only real-life data, including exclusively the results of the final clinical reports. To the best of our knowledge, this is the first study to show a deep analysis of PAEs in BMA DCC. This is also the first study to include a comprehensive and evidence-based approach comparison between BMCS DCC and IP FDC. Limitations of our study include the different sample sizes according to the type of cell in the IP FCD, but the aim of the study was to include only results of the final clinical report. The relatively small sample size of the control group may also be considered a potential limitation. Despite recent innovations in the field, cytomorphological DCC remains the most efficient starting point for a BMA-based diagnostic approach. Larger studies focusing on specific entities and target cutoffs could help to further know the limits of the cytomorphological performance.

In the clinical laboratory scenario, quality management is mandatory. ISO 15189 accreditation helps to reach and maintain specific technical and management standards that guarantee a high level of quality as well as a circle of continuous improvement. The clinical laboratory is also evolving. Human resources are increasingly scarce as the workload continues to grow. Adapting to a continuously changing environment requires combining quality assurance with efficiency. Manual techniques, such as BMA DCC, are time-consuming. Moreover, the access to medical records, the comparison with previous studies, the attention to quality issues, the coordination with other staff members, the teaching of hematologist trainees, and many other tasks, make time management essential for optimal performance.

## 5. Conclusions

The BMCS DCC remains a key study in the current hematological diagnostic workup, together with the IP FDC. Preanalytical issues play a critical role in BMA samples. Any PAE should be recorded in the final BMCS DCC report. PAEs should be standardized according with the SOP and monitored according with the ISO 15189 Standard. Our evidence-based comparison study between 200- and 500-cell BMCCS DCC, as well as between BMCS (both cutoffs) and IP FDC, show a lack of statistically or clinically meaningful differences for all type of cells. Our data suggest that using a minimum of 200-cell DCC along with pre- and post-count evaluations is safe and efficient for the vast majority of studies in daily clinical practice. A higher DCC is proposed, whenever possible, in cases in which the percentage of the cell of interest is near of an established cutoff. Our evidence-based approach guarantees both the highest level of quality and the expected optimization of resources. Therefore, we recommended it to be used in every clinical laboratory aiming to get a comprehensive, integral, and efficient hematological diagnosis, particularly in those accredited by the ISO 15,189 Standard. In the era of the artificial intelligence, cytomorphology still remains a key diagnostic step in the hematological diagnosis and every effort should be made to balance both safety and efficiency.

## Figures and Tables

**Figure 1 diagnostics-13-01071-f001:**
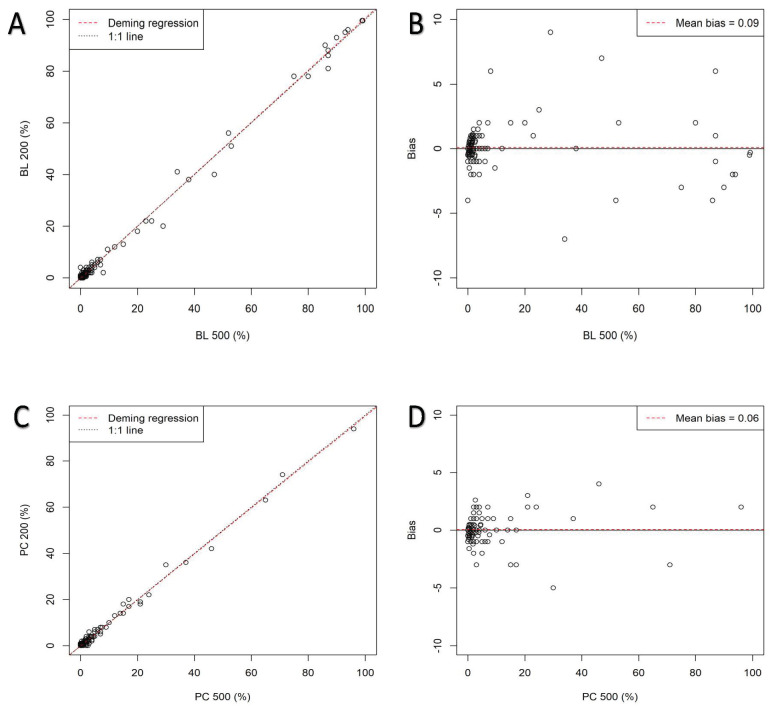
Deming linear regression and difference bias plots for blasts (**A**,**B**) and plasma cells (**C**,**D**).

**Table 1 diagnostics-13-01071-t001:** Recommendations for bone marrow aspirate differential cell count according with the level of evidence.

Reference	Year	Suggested DCC-Count	Level of Evidence
Jandl, J.H. [4]	1996	200 (X2)	EO
Osorio, G. [5]	1996	200–500	EO
Perkins, S.L. [6]	2004	500	EO
Woessner, S. & Florensa, L. [7]	2006	500–1000	EO
Lee, S.H. et al. [8]	2008	300–500	EO
Theil, K.S. [9]	2012	200–500	EO
Ryan, D.H. [10]	2016	300–500	EO
Keohane, E.M. et al. [11]	2016	300–1000	EO
Swerdlow, S.H. et al. [12]	2017	500	EO
Bain, B.J. et al. [13]	2017	200–500	EO
Abdulrahman, A.A. et al. [14]	2018	300	EBA

Abbreviations: DCC: differential cell count; EBA: Evidence-based approach comparison study; EO: Expert Opinion (Individual or Group).

**Table 2 diagnostics-13-01071-t002:** Characteristics of the samples.

Samples Characteristics	Training Set	Validation Set	Complete Set	Control Group
*n*	330	330	660	122
Group *n* (%) A B	287 43 (13%)	263 67 (20.3%)	550 110 (16.7%)	110 12 (9.8%)
Type of Study (Group A) *n* (%) New Reevaluation	113 174 (60.6%)	103 160 (60.8%)	216 334 (60.7%)	44 78 (63.9%)
Pediatric Samples *n* (%)	39 (11.8%)	42 (12.7%)	81 (12.3%)	7 (5.7%)

**Table 3 diagnostics-13-01071-t003:** Preanalytical events.

Preanalytic Event	Complete Set *n* (%)	Control Group *n* (%)	*p* Value
Clotted Sample	36 (5.5)	1(0.8)	0.027
Any Other Pae	128 (19.4)	22 (18.0)	NS
Platelet Aggregates	99 (15)	ND	-
Pb Contamination	5 (0.8)	15 (12.3)	<0.000
Low Volume	7 (1.1)	7 (5.7)	<0.000
>24 h-Delay Reception	3 (0.5)	0 (0)	NS
No Clinical Information	8 (1.2)	0 (0)	NS
Duplicated Sample	1 (0.2)	0 (0)	NS
Sample Not Received	1 (0.2)	0 (0)	NS
Wrong Delivered Sample	1 (0.2)	0 (0)	NS
Sample Without Analytical Request	2 (0.3)	0 (0)	NS
Wrong Analytical Request	1 (0.2)	0 (0)	NS

Abbreviations: ND: not determined; NS: no significance.

**Table 4 diagnostics-13-01071-t004:** Comparison analysis between BMCS 200-cell versus 500-cell DCC and between BMCS and IP according to main types of cells *.

	BMCS 200- vs. 500-DCC		BMCS (Complete Set) vs. IP	
Cell Type	Complete Set	Control Group	200-Cell DCC	500-Cell DCC
Blast	0.997	0.997	0.929	0.947
Promyelocyte	0.981	0.862	0.977	0.983
Erythroblast	0.978	0.953	0.673	0.682
Lymphocyte	0.963	0.973	0.768	0.806
Plasma cell	0.994	0.991	0.959	0.958
Monocyte	0.957	0.845	0.814	0.827

* All associates *p* values are significant at <0.000.

**Table 5 diagnostics-13-01071-t005:** Comparison analysis between BMCS 200-cell and 500-cell DCC results for the complete set (*n* = 550) according to cell type, using Deming Linear Regression.

Cell Type	Slope, CI 95%	Intercept, CI 95%	*r* Value	Mean Bias
Blast	1.01 (0.98–1.02)	−0.11 (−0.18, −0.05)	0.997	0.09
Promyelocyte	1.03 (1.01, 1.40)	−0.04 (−0.48, 0.02)	0.981	0.00
Erythroblast	1.02 (1.01, 1.04)	−0.91 (−1.37, −0.42)	0.978	0.25
Lymphocyte	1.00 (0.97, 1.07)	−0.10 (−0.43, 0.11)	0.963	0.07
Plasma cell	1.00 (0.97, 1.04)	−0.05 (−0.11, 0.00)	0.994	0.06
Monocyte	1.00 (0.92, 1.10)	0.13 (−0.21, 0.45)	0.957	−0.15

**Table 6 diagnostics-13-01071-t006:** Diagnostic performance of 200-cell DCC at the 20% blast cutoff.

Training Set	500 RDC			
200 RDC	≥20%	<20%	Total	
≥20	9	0	9	100% PPV
<20	0	278	278	100% NPV
Total	9	278	287	
	100% Se	100% Sp		
**Confirmation Set**	**500 RDC**			
200 RDC	≥20%	<20%	Total	
≥20	10	0	10	100% PPV
<20	1	252	253	99.6% NPV
Total	11	252	263	
	90.9% Se	100% Sp		
**Complete Set**	**500 RDC**			
200 RDC	≥20%	<20%	Total	
≥20	19	0	19	100% PPV
<20	1	530	531	99.8% NPV
Total	20	530	550	
	95% Se	100% Sp		

DCC, differential cell count; NPV, negative predictive value; PPV, positive predictive value; Se, sensitivity; Sp, specificity.

**Table 7 diagnostics-13-01071-t007:** Diagnostic performance of 200-cell DCC at the 10% blast cutoff.

Training Set	500 RDC			
200 RDC	≥10%	<10%	Total	
≥10	9	1	10	90% PPV
<10	0	277	277	100% NPV
Total	9	278	287	
	100% Se	99.6% Sp		
**Confirmation Set**	**500 RDC**			
200 RDC	≥10%	<10%	Total	
≥10	13	0	13	100% PPV
<10	0	250	250	100% NPV
Total	13	250	263	
	100% Se	100% Sp		
**Complete Set**	**500 RDC**			
200 RDC	≥10%	<10%	Total	
≥10	22	1	23	95.7% PPV
<10	0	527	527	100% NPV
Total	22	528	550	
	100% Se	99.8% Sp		

DCC, differential cell count; NPV, negative predictive value; PPV, positive predictive value; Se, sensitivity; Sp, specificity.

**Table 8 diagnostics-13-01071-t008:** Diagnostic performance of 200-cell DCC at the 5% blast cutoff.

Training Set	500 RDC			
200 RDC	≥5%	<5%	Total	
≥5	15	2	17	88.2% PPV
<5	1	269	270	99.6% NPV
Total	16	271	287	
	93.8% Se	99.3% Sp		
**Confirmation Set**	**500 RDC**			
200 RDC	≥5%	<5%	Total	
≥5	13	0	13	100% PPV
<5	1	249	250	99.6% NPV
Total	14	249	263	
	92.9% Se	100% Sp		
**Complete Set**	**500 RDC**			
200 RDC	≥5%	<5%	Total	
≥5	28	2	30	93.3% PPV
<5	2	518	520	99.6% NPV
Total	30	520	550	
	93.3% Se	99.6% Sp		

DCC, differential cell count; NPV, negative predictive value; PPV, positive predictive value; Se, sensitivity; Sp, specificity.

## Data Availability

Not applicable.

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
