# Peer review of "Cell Count Differentials by Cytomorphology and Next-Generation Flow Cytometry in Bone Marrow Aspirate: An Evidence-Based Approach"

_diagnostics, 2023, doi:10.3390/diagnostics13061071_

Round 1

Reviewer 1 Report

1.      Please mention your study limits and suggest some future research topics

2.      In References, the sources are written in different styles. Please update the reference list.  It is necessary to bring in accordance with the requirements of the journal for the design of References. If possible, indicate DOI.

3.      Please use some innovative keywords.

4.      Please mention your study limits in the abstract.

5.      The Conclusions should reflect what the practical application of the results obtained in this study is. In what climatic conditions should the recommendations of the authors be taken into account?

Author Response

Comments and Suggestions for Authors

  1. Please mention your study limits and suggest some future research topics

Some limitations of the study were indicated in the penultimate paragraph of the discussion (line 325). We have added another limitation, the relatively small sample size of the control group. We also suggest future research avenues in the field (highlighted paragraph).

  1. In References, the sources are written in different styles. Please update the reference list.  It is necessary to bring in accordance with the requirements of the journal for the design of References. If possible, indicate DOI.

References have been updated and DOI have been added when it was available.

  1. Please use some innovative keywords.

New keywords have been added.

  1. Please mention your study limits in the abstract.

Several limitations have been highlighted in the abstract.

  1. The Conclusions should reflect what the practical application of the results obtained in this study is. In what climatic conditions should the recommendations of the authors be taken into account?

This recommendation has been added in the last sentence of the conclusions.

Thank you.

Reviewer 2 Report

Well written manuscript, however the ISO standard demanded needs for optimization. 

1. What is the main question addressed by the research?  Ans.: The main question addressed by the manuscript is about deep analysis of PAEs in BMA DCC.  2. Is it relevant and interesting?  Ans.: Yes its relevant and interesting unless the authors are requesting for ISO 15189 Standard for PAEs record keeping.  3. How original is the topic?  Ans.: Yes the topic is original as seen from their pre clinical and post clinical data evaluation.  4. What does it add to the subject area compared with other published material?  Ans.: The comparative analysis part is not addressed much by the authors as they wish for ISO 15189 Standard for BMCS DCC report analysis. It requires a higher rate of literature reviews.  5. Is the paper well written? Ans.: Yes the manuscript is well written.  6. Is the text clear and easy to read?  Ans.: Yes the text is clear and easy to understand.  7. Are the conclusions consistent with the evidence and arguments presented?  Ans.: yes the conclusion is consistent with the evidence and discussion. Here they proposed the real-life data analysis of PAEs in BMA DCC via deep analysis methodology.   8. Do they address the main question posed? Ans.: Yes they addressed the main question posted. 

Author Response

Comments and Suggestions for Authors

Well written manuscript, however the ISO standard demanded needs for optimization. 

  1. What is the main question addressed by the research? Ans.: The main question addressed by the manuscript is about deep analysis of PAEs in BMA DCC. We thanks the reviewer's words.
  2. Is it relevant and interesting? Ans.: Yes its relevant and interesting unless the authors are requesting for ISO 15189 Standard for PAEs record keeping. We recognize that the engagement with the ISO15189 Standard is a constant opportunity to improve our daily clinical practice.
  3. How original is the topic? Ans.: Yes the topic is original as seen from their pre clinical and post clinical data evaluation.  We thanks the reviewer's words.
  4. What does it add to the subject area compared with other published material? Ans.: The comparative analysis part is not addressed much by the authors as they wish for ISO 15189 Standard for BMCS DCC report analysis. It requires a higher rate of literature reviews. We agree with the reviewer. We have made a deep literature search on this topic but there is a remarkable lack  of evidence-based works on this topic. Our aim was to find a balance between quality and efficiency from a practical point of view.
  5. Is the paper well written?Ans.: Yes the manuscript is well written. We thanks the reviewer's words.
  6. Is the text clear and easy to read? Ans.: Yes the text is clear and easy to understand.  We thanks the reviewer's words.
  7. Are the conclusions consistent with the evidence and arguments presented? Ans.: yes the conclusion is consistent with the evidence and discussion. Here they proposed the real-life data analysis of PAEs in BMA DCC via deep analysis methodology.  We thanks the reviewer's words.
  8. Do they address the main question posed?Ans.: Yes they addressed the main question posted. We thanks the reviewer's words.

Thank you for your kind consideration

Reviewer 3 Report

The paper by Rafael Ríos-Tamayo et al is a text that has already been properly revised. It describes a comparative study between the analysis of 200 vs 500 events acquired in laboratory analysis of haematological diseases at diagnosis or during the follow-up.

The sections of the text are properly described. 

The authors should clearly indicate the benefits of this method both in the part of the results and in the conclusions.

Author Response

We appreciate the reviewer's comments. We agree that the benefit of this method should be more clearly emphasized. We have added a comment about it in the two indicated sections.